# Health promotion intervention to prevent risk factors of chronic diseases: Protocol for a cluster randomized controlled trial among adolescents in school settings of Chandigarh (India)

**Sandeep Kaur[1], Manmeet Kaur[1]\*, Rajesh Kumar[1,2,3]**

1 Department of Community Medicine and School of Public Health, Post-Graduate Institute of Medical Education and Research, Chandigarh, India, 2 Department of Epidemiology and Population Health, London School of Hygiene and Tropical Medicine, London, United Kingdom, 3 School of Public Health and Community Medicine, University of New South Wales, Sydney, Australia

\* mini.manmeet@gmail.com

**Funding:** Partial funding was awarded to MK with project ID 5562 for undertaking laboratory tests, from The George Institute for Global Health,

## Abstract

### Background

Chronic diseases like diabetes, cardiovascular diseases and cancers are on the rise. Most of the risk factors of these diseases commence in Adolescence. Therefore, a cluster randomised controlled trial is designed to evaluate the effect of school-based health promotion intervention on the risk factors of chronic diseases.

### Methodology

Considering school as a cluster, twelve schools will be randomly selected from the public schools of Chandigarh, a city in India. After baseline assessment, six schools will be randomly allocated to intervention and six to the control arm. Study participants will be students of 8th grade (age 10–16 years), their parents and teachers. A sample of 360 students (12 clusters x 30 students) has been estimated to provide statistically valid inference. The PRECEDE PROCEED Model will be used to develop health promotion interventions to prevent the use of an unbalanced diet, physical inactivity, alcohol, and tobacco. Interventions will be implemented for six-months in the school setting. For students, the intervention will comprise interactive learning sessions of 30 minutes duration per week and physical activity sessions of 30 minutes duration four times every week. Educational sessions will be conducted for parents and teachers for 30 minutes, four times during the intervention period. Primary outcomes will be changes in the prevalence of behavioural risk factors from pre- to post-intervention. Changes in anthropometric, physiological, and biochemical measures will be the secondary outcomes. The difference-in-difference (DID) method will be used to measure the net change in the outcomes.

Hyderabad, India. The funders had no role in the larger study design, data collection and analysis, decision to publish, or preparation of the manuscript.

**Competing interests:** The authors have declared that no competing interests exist.

## Discussion

It is essential to understand whether health promotion interventions implemented in the school setting simultaneously targeting adolescents, teachers, and parents are effective. Using the PRECEDE-PROCEED model for planning, implementing, and evaluating the intervention as part of a cluster Randomized Controlled Trial design with DID analysis, could objectively assess the impact.

## Introduction

Technological advancements have improved the living conditions of people all over the world. However, this transition has shifted dietary behaviours from traditional home-cooked meals to processed food high in sugar, salt, and fat and low in fruits and vegetables. It has also increased the sedentary behaviours of individuals. Moreover, the growing availability and early exposures have led to increased tobacco and alcohol-related products early in life [1, 2].

Unbalanced diets, physical inactivity and tobacco and alcohol use, are the behavioural risk factors leading to chronic diseases. Currently, about 50% of total deaths globally are caused by chronic diseases. Low- and middle-income countries like India face a similar situation wherein 40% of all deaths are caused by chronic diseases [3]. Many of these deaths occur at a young age in India, leading to further loss of potentially productive life years [4].

Several health promotion interventions consisting of behaviour change strategies and activities have been recommended, and these are being implemented to reduce risk factors for chronic diseases [5]. However, most intervention trials have been conducted in high-income countries [6–8] and few in the low- and middle- income countries [9–13]. The trials carried in high-income countries may not be effective in low- and middle- income countries, specially India, as interventions need to be developed considering diverse cultural, socio-demographic and economic factors [14]. Most of the studies carried out in low- and middle-income countries in the school settings among adolescents have either focussed on one of the risk factors [9, 10] or have implemented interventions of shorter duration [11, 12]. Most of these studies also lack any theoretical basis for the intervention development and implementation [15, 16]. Evidence suggests that interventions should be based on behaviour change theories or models to be more effective in the targeted populations [17–19].

As chronic diseases of adulthood result from behavioural risk factors that usually begin during early Adolescence [20–22], interventions should target adolescents. However, only a few randomised controlled trials (RCT) have been carried out among adolescents in the school settings of Low-middle Income countries [9, 12]. Moreover, none of these studies have included parents and teachers even though both school and home environments may influence adolescents [9, 12]. Hence, we have designed a study to evaluate the effectiveness of a school-based health promotion intervention package in improving behavioural risk factors among adolescents, their teachers, and parents.

The null hypothesis of the study is: School-based health promotion intervention will not lead to any change in dietary intake of salt (g/day), sugar (g/day), fruits (g/day) and vegetables (g/day), physical inactivity (%), tobacco and alcohol use (%), anthropometric (height, weight, waist circumference, hip circumference, mid-upper arm circumference, subscapular and triceps skinfold thickness), physiological (blood pressure) and biochemical (fasting plasma glucose and urinary sodium excretion levels) measures among adolescents.

The alternate hypothesis of the study is: School-based health promotion intervention will lead to change in dietary intake of salt (g/day), sugar (g/day), fruits (g/day) and vegetables (g/day), physical inactivity (%), tobacco and alcohol use (%), anthropometric (height, weight, waist circumference, hip circumference, mid-upper arm circumference, subscapular and triceps skinfold thickness), physiological (blood pressure) and biochemical (fasting plasma glucose and urinary sodium excretion levels) measures among adolescents.

## Materials and methods

### Study area

The study will be carried out in public schools of Chandigarh city in India. Chandigarh is a Union Territory (UT) and capital of two states—Punjab and Haryana. It has a total population of around one million. Out of 188 schools, 115 are public, and 73 are private [23]. The school timings are from 8 AM to 2 PM in most public schools, with each subject period of 35 minutes duration. Most schools have a 'zero period' in a day, which is free time for adolescents. The 'zero periods' are a good time for intervention implementation without disturbing the overall school routine of the adolescents.

### Study design

A cluster randomised controlled trial (cRCT) design will be used in which 12 schools will be randomly selected from the list of 115 public schools. The randomly selected twelve schools (clusters) will be stratified based on the household income of the study participants. Using stratified random sampling equal number of clusters from each of the income strata will be randomly allocated to the intervention and control arms. This stratified random sampling will be employed to balance the potential confounders and reduce between-cluster variance for estimating the true effect of the intervention [24]. The second author (MK) will conduct randomisation procedures to assign the schools and participants to the intervention and control arms. Whereas, First Author (SK) will enrol the participants.

Formative research will also be carried out to understand the social environment and other contextual factors. Study design features, participant's enrolment, intervention period, and study variables at baseline and end-line analysis are summarised in Fig 1. The study was registered under the Clinical Trial Registry of India (CTRI/2019/09/021452; 30/09/2019).

### Study participants

The study participants will include school students of 8th grade (age range 10–16 years), their parents and teachers. The purpose of selecting adolescents in this age group is to catch them 'young', when behavioural risk factors of chronic diseases start to emerge [18]. Either of the parents (mother or father) or guardian will be recruited into the study based on adolescent choice, parents' consent and their time availability. The teachers who have direct interaction with study adolescents for a more extended time, such as class teachers, subject teachers, physical education teachers, counsellors, or the person in charge of the mid-day meal or medical assistant, will also be selected for the study.

### Inclusion and exclusion criteria

Public schools of UT Chandigarh will be eligible for participation in the study. Consenting participants residing in the study area and not intending to leave that area in the next year from the time of recruitment will be included in the study. Only adolescents studying in 8th grade and their parents and teachers below 65 years will be enrolled.

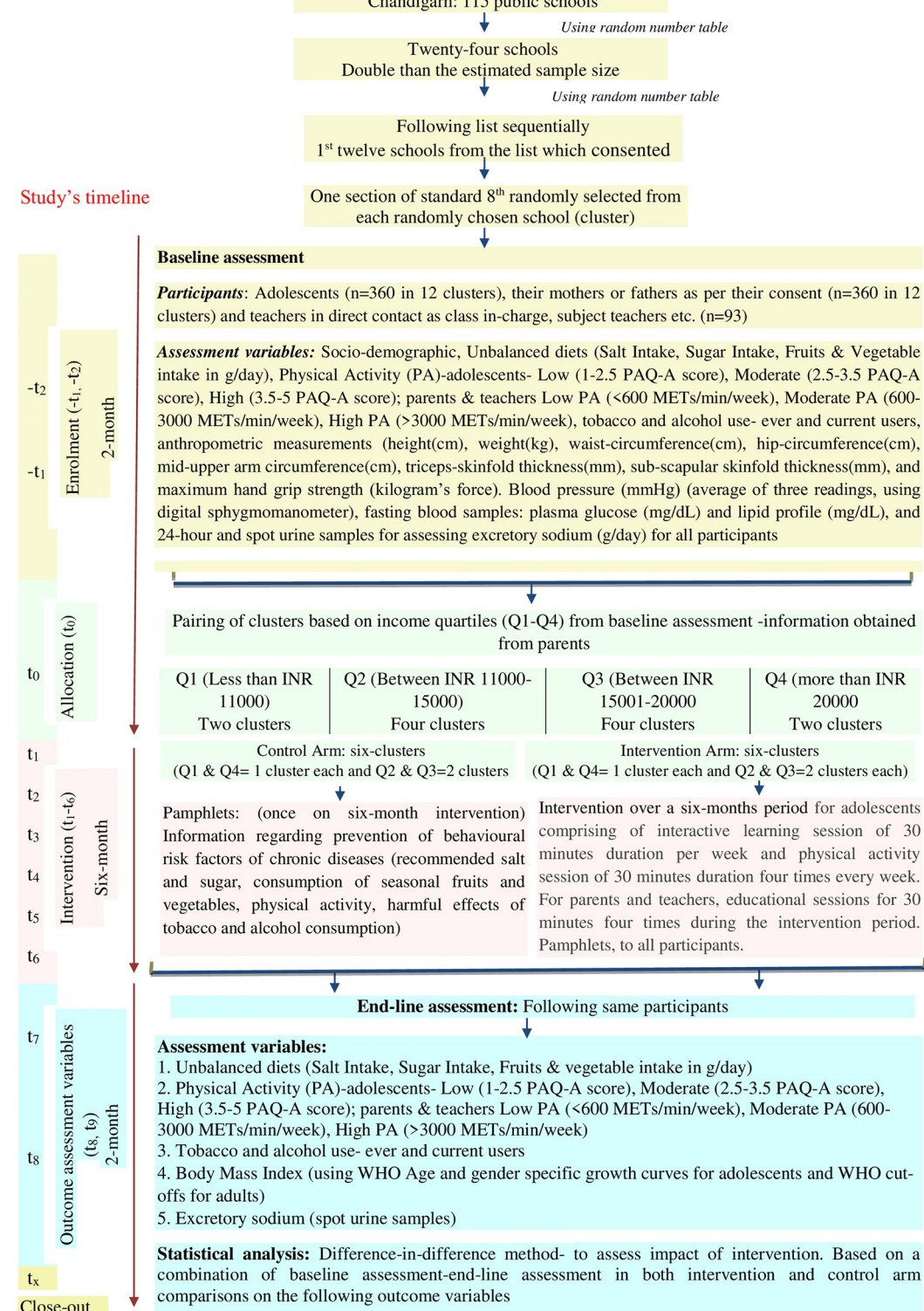

**Fig 1. Study flow design.**

## Sample size

A total of 12 clusters (schools) are sampled based on the formula mentioned below:

$$\mathbf{c} = \mathbf{1} + \mathbf{f}[\boldsymbol{\pi_0}(\mathbf{1} - \boldsymbol{\pi_0})/\mathbf{m} + \boldsymbol{\pi_1}(\mathbf{1} - \boldsymbol{\pi_1})/\mathbf{m} + \mathbf{k}^2(\boldsymbol{\pi_0^2} + \boldsymbol{\pi_1^2})]/(\boldsymbol{\pi_0} - \boldsymbol{\pi_1})^2$$

Where $c$ is the number of clusters required, $f$ = 7.84 for 5% type I error and 80% power, $\pi_1$ is the expected proportion of the risk factor in the intervention arm, $\pi_0$ is the expected proportion of risk factor in the control arm, $m$ is the number of individuals in each cluster (assumed equal in all clusters), and $k$ is the coefficient of variation in the (true) rates or proportions between clusters in each treatment arm. As most of the schools have a class strength of about 30 students in 8[th] grade, the size of the cluster ($m$) is taken to be 30 students.

The $c$ value for each primary outcome is calculated separately using data from the previous studies, as shown in Table 1 [25–29]. The maximum $c$ value, out of all four risk indicators, is considered the sample size for the study.

Thus, a sample size of 360 students (12 clusters x 30 students per cluster); 180 each in the intervention and control arm is estimated to provide valid statistical inference about the impact of the intervention. As one of the selected student's parents will also participate in the study, the sample size for parents will also be 360. Lastly, it is observed that on average about seven teachers are in direct interaction with the students of 8[th] grade; therefore, the sample size of teachers will be 84 (12 clusters x 7 teachers).

For the formative research, about ten Focus Group Discussions (FGDs) will be carried out with participants from the study area. Out of ten FGDs, four will be with the students (2 male, two female), four with their parents (2 fathers, two mothers), and two with teachers (1 male, one female). Each of the FGDs will include a minimum of 5 and a maximum of 8 participants.

## Sampling method

Twenty-four schools (double the required number of twelve schools) will be randomly chosen by using a random number table from the list of 115 public schools in Chandigarh city [30]. Administrators of these 24 schools will be approached from the list sequentially for obtaining their consent. For each refusal, the next school on the list will be approached. After receiving approval from the first twelve schools, the list will be closed. From within the 12 selected schools, 12 clusters will be selected randomly using the following procedure.

As each school has four eighth grade sections, namely, A, B, C and D, a number will be assigned on folded paper slips for all the sections of the 8[th]-grade class. Later, a slip will be randomly selected from the shuffled lot by a person other than the researcher. The section number in the slip will then be selected as the cluster for the study. The method mentioned above will be applied to choose a section of 8th grade in each of the selected twelve schools. Adolescents studying in 8[th] grade are between 10–16 years of age. It is known from the literature that; six-month intervention is sufficient to bring out the desired behaviour change. But the

**Table 1. Sample size estimation based on the prevalence of risk factors of chronic diseases among adolescents.**

| Risk Factor | $\pi_0$ | $\pi_1$ | $K$ | $C$ |
|---|---|---|---|---|
| Inadequate fruits and vegetable intake | 0.852 | 0.596 | 0.025 | 12.2 |
| High salt intake | 0.223 | 0.162 | 0.094 | 12.1 |
| High sugar intake | 0.48 | 0.34 | 0.041 | 12.5 |
| Physical inactivity | 0.232 | 0.162 | 0.099 | 12.1 |
| Tobacco use | 0.049 | 0.034 | 0.163 | 11.4 |
| Alcohol use | 0.345 | 0.242 | 0.057 | 12.4 |

biochemical measures are not impacted, which has been the basis for developing the study's outcomes [31, 32].

## Outcomes

The primary outcomes of the study will be the change in dietary intake (g/day) of sugar, salt, fruits and vegetables, physical activity (metabolic equivalent/minute/week), current tobacco and alcohol use (%) among adolescents, pre-and post- health promotion intervention implementation in both the intervention and control arms.

Secondary outcomes will include changes in body mass index (kg/m$^2$) blood pressure (mmHg), and urinary sodium excretion (mg/day)) among adolescents of both intervention and control arms before and after the delivery of the intervention package.

## Intervention development

The intervention is based on the PRECEDE-PROCEED model (PPM) to increase the likelihood of effectiveness [33, 34]. This model will inform the development, implementation, and evaluation of the health promotion intervention as described in the Fig 2. The intervention design will be developed by systematically using various phases of the PPM, as described below.

**Phase 1 social diagnosis.** Formative research will be conducted to understand participants' social environment and their level of awareness regarding behavioural risk factors for chronic diseases.

**Phase 2 epidemiological and behavioural diagnosis.** Epidemiological diagnosis will focus on specific health issues like the prevalence of chronic diseases, and behavioural diagnosis will deal with risk factors such as unbalanced diet, physical inactivity, tobacco and alcohol use. These factors will be assessed through a cross-sectional survey. The end-line survey will be carried in both arms using the same methods, tools, apparatus and following the same study participants as the cross-sectional survey. So, this cross-sectional survey will also form the baseline for the present study.

**Phase 3 educational and ecological diagnosis.** In this phase, social and environmental factors leading to specific behaviours will be identified from the data collected through formative research and baseline assessment. The prevalence of behavioural risk factors among the participants from baseline data will help frame the objectives and inform the intervention development.

**Phase 4 intervention alignment.** This phase will involve consultation with adolescents, parents, and teachers to develop the intervention. To strengthen the intervention, findings from focus group discussions will be utilised to understand 'participants' requirements and suggestions regarding the intervention. Additionally, a consultation workshop will be held, where teachers of all the selected schools will participate. The study objectives and a draft of the intervention plan will be presented to them, and their suggestions for improving the intervention plan will be sought. An intervention module for both adolescents, parents and teachers will be prepared.

The intervention package will consist of:

1. Interactive Learning Sessions: In the classroom settings, lecture and discussion sessions will be conducted with all participants on behavioural risk factors of chronic diseases. These will be focused on primarily the importance of a balanced diet, especially about the intake of adequate amount of seasonal fruits and vegetables, and recommended levels of salt and sugar as per WHO guidelines, ways to measure salt and sugar in home-cooked food using measuring tea-spoons, calculating amounts of salt and sugar in processed foods by reading

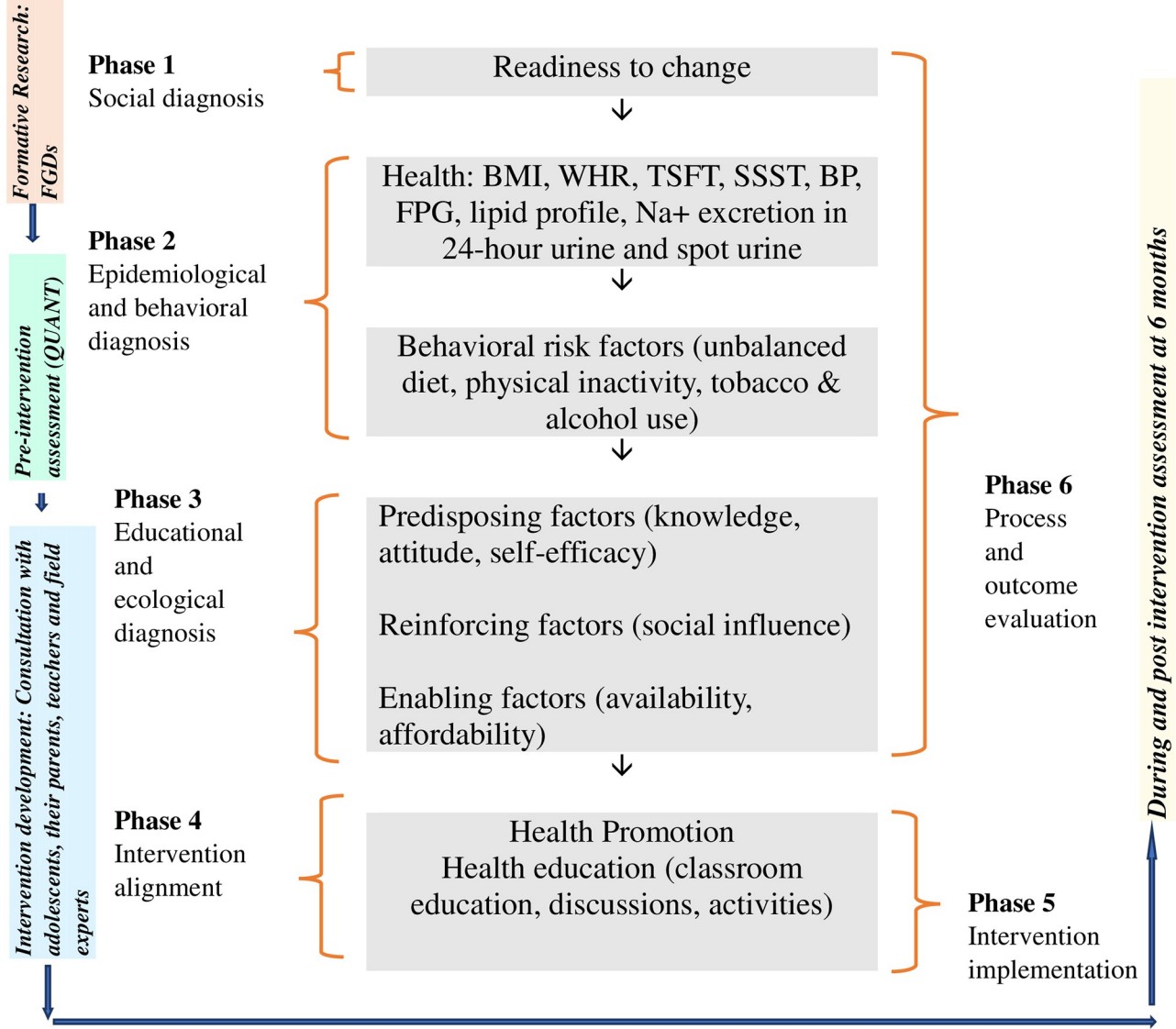

**Fig 2. PRECEDE-PROCEED model adapted from Green and Kreuter (1999) [34].**

labels, harmful effects of tobacco and alcohol use, the importance of physical activity, and secondarily on the ways to manage hypertension and diabetes including adherence to medicine. Activities such as poster making for a balanced diet debates on processed food, salt and sugar consumption, and tobacco and alcohol use will also be conducted.

2. Physical Activity Sessions: Students will participate in physical activities such as sports and athletics under the supervision of a physical training instructor for 30 minutes 4 times per week.

3. Peer to Peer Sessions: Students will be encouraged to share their experiences of avoiding tobacco and alcohol use [35].

Besides this, a pamphlet containing information on chronic diseases' risk factors will be given to all study participants. As outlined in Table 2, a 'Behaviour Change 'Communication'

**Table 2. Behaviour change communication matrix for health promotion interventions to reduce chronic disease risk factors in public schools in Chandigarh, India.**

| Communication objectives | Audience | Barriers | Opportunities | Activities | Person responsible | Frequency & time in one cluster (six month) | Channel of communication |
|---|---|---|---|---|---|---|---|
| • To promote the recommended levels of salt, sugar, fruits and vegetable intake, and physical activity and reduction in the use of tobacco and alcohol | Teachers: Teachers in direct interaction with randomly selected clusters (sections of 8th grade) | Different class schedules of the teachers | School hours are the best time to interact with teachers in groups | Educational Sessions with teachers keeping communication objectives in focus | Researcher | Four sessions of thirty minutes during intervention period | Verbal (2-way communication) |
| | Parents of adolescents from randomly selected clusters | Not all parents are available at the time of the parents-teachers meeting in schools | • Utilising home environment for promoting healthy diets by educating parents (as mostly, mothers are the food makers of the house) on the harmful effects of excessive intake of salt & sugar and less consumption of fruits and vegetables<br><br>• The Parents Teacher Association (PTA) meetings may be used to interact with parents in school settings | Educational Sessions with parents keeping communication objectives in focus | Researcher | Four sessions of thirty minutes during intervention period | Verbal (2-way communication) |
| • To manage hypertension and diabetes by discussing the importance of medicine adherence | *Primary audience*: Adolescents of the randomly selected clusters (sections of 8th grade) | • Vendors outside school premises | • To catch adolescents before risky behaviours start developing | Educational classes and classroom discussions | Researcher | 30 minutes session every week | Verbal (2-way communication) & Digital audio-visual method (PowerPoint presentation using the laptop) |
| | *Secondary audience*: Parents & teachers of adolescents | • Unhealthy food options in the school canteens | • Using schools as a platform for promoting healthy behaviours | Physical activity sessions | Researcher/ physical education teachers | Four sessions every week with each session of 30 minutes | Verbal (2-way communication) |
| | | • Study load (tuitions after school hours & homework) | | Other interactive activities: Poster making competition & grow your own herbs | Researcher | Two sessions during intervention period with each session of about 45 minutes | Verbal (2-way communication) |
| | | • Stringent school time-table | | Peer-to-peer education for reduction in the use of tobacco and alcohol | Researcher | One session every month with each session of 15 minutes | Verbal (2-way communication) |

(BCC) Matrix will be prepared for deciding on the messages, channels, modes of communication, and indicators for change before implementing the intervention. The channel of communication will be inter-personal two-way communication, including audio-visual and digital communication. This matrix will be one of the crucial highlights of the present study as most of the earlier studies lacked this kind of micro-planning before implementing the intervention in the school settings [36].

**Phase 5—Intervention implementation.**   *Intervention arm.* The intervention will focus on building adolescents' self-efficacy at the micro-level and that of parents and teachers at the meso level.

Intervention implementation strategies, presented below, will be based on three components: Capability, Opportunity and Motivation as per the Behaviour Change Wheel [5].

1. Parents and teachers are involved in providing a supportive environment for adolescents at home and at school to promote positive behaviour change early in life.

2. Creating awareness by providing information on healthy dietary habits, the importance of physical activity, and the harmful effects of tobacco and alcohol use.

3. Promotion of consumption of less expensive and more nutritious seasonal fruits and vegetables.

4. Avoiding food high in sugar and salt content and refraining from tobacco and alcohol use.

5. Substituting snacks with fruit and vegetable salads.

6. Motivating the participants in reducing salt and sugar consumption by using measuring spoons while cooking as per the WHO recommendations.

7. Promoting kitchen gardening encourages participants to grow herbs and spices in earthen pots as herbs can reduce salt usage. Herbs and spices also add flavours and enhance the taste of food in places of salt.

8. Highlighting potential cost savings for individuals by creating awareness among participants regarding the benefits of eating more fruits and vegetables, eating less sugar and salt, increasing physical activity, and reducing or stopping the use of tobacco and alcohol to prevent chronic diseases, which in turn will help to reduce medical expenses related to chronic diseases.

9. Providing opportunities for playground activities in school to increase the level of physical activity among adolescents.

10. Encouraging peer to peer learning for motivation and support for not using tobacco and alcohol.

A micro plan will be prepared to ensure intervention adherence. It includes providing measuring tea-spoons of 5g to all the participants to promote the recommended amount of salt and sugar in their daily routine. For physical activity, four outdoor game sessions of 30 minutes each will be carried with the help of the respective physical education teachers in all six schools every week. For tobacco and alcohol, peer-to-peer sessions will also be held once every month to create awareness among adolescents regarding the harmful effects of alcohol and tobacco consumption.

Interventions will be implemented by the researcher (SK) in a school setting over six months. For adolescents, interventions will comprise interactive learning sessions of 30 minutes duration per week and a physical activity session of 30 minutes four times per week. Educational sessions will be conducted for parents and teachers for 30 minutes four times during

the intervention period. Besides the lecture and discussion, at least fifty percent of the time will be allocated to poster making, debates, and experience sharing.

*Control arm*. Participants of the control arm will be provided with a pamphlet in the local language so that all study participants, especially parents, can understand the information easily. This pamphlet will be based on the dietary recommendations of the National Institute of Nutrition (NIN), India [37]; the levels of physical activity for both adolescents and adults, salt and sugar consumption, and consequences of tobacco and alcohol as per World Health Organization recommendations [38–42]. Study participants will be requested to read the information in the pamphlet in their own time and convey the same information to their family members. The same pamphlet will also be provided to the participants in the intervention arm.

**Phase 6 process and impact evaluation.** Process evaluation to be initiated at the time of intervention implementation to assess whether the intervention is being implemented as per the plan, to analyse the factors facilitating and hindering the use of the health promotion intervention, and for identifying processes that require improvement. Information regarding the uptake of intervention, changes incorporated in behaviours, the number of adolescents following the behaviours and the number of them conveying the same to their family members will be collected in writing. This information will be recorded by interviewing the adolescents once in the intervention arm.

Outcome evaluation will be done after six months to examine the impact of the intervention on the outcome indicators. The Difference-in-Differences (DID) method will be used to assess net changes after implementing the health promotion intervention in both the control and the intervention arm [43, 44].

## Data collection

The dietary behaviour assessment will be through the Multiple-Pass Method by using two 24-hour dietary recalls on non-consecutive days during weekdays. After a thorough explanation of the dietary tool, all the adolescents in the same cluster will fill their forms themselves (self-administration). Whereas in-person interviews will be carried out with parents and teachers. The first 24-hour dietary recall collection will be during the first interaction with all the participants in the schools. The second 24-hour dietary recall collection will be a week later. The researcher will carry out all the interviews with parents and teachers and explain the tool to adolescents [45].

Data collected through the 24-hour dietary recall will be analysed using the PURE study software [46]. This software uses National Institute of Nutrition (NIN) dietary guidelines for Indians to calculate the quantity of various macro and micronutrients from the quantity of various food items consumed by a person in a day. Using this software daily intake of sugar (g), salt (g), fruits (g), and vegetable (g) will be estimated. The participants' nutrient intake will also be compared against the recommended dietary allowances according to NIN dietary guidelines [35]. Validated physical activity tools based on a seven-day recall period will be used. Physical Activity Questionnaire for Adolescents (PAQ-A) will be used, and for parents and teachers, Short International Physical Activity Questionnaire for Adults (IPAQ) will be used [47, 48]. The data on adults' physical activity level will be converted into Metabolic equivalent (MET) using the manual of IPAQ [48]. Physical activity will be further categorised using the scale provided in the user manual [48]. For adolescents, the PAQ-A user manual will be used to compute a composite physical activity score. It will be categorised into mild, moderate and high according to the scale provided in the PAQ-A manual [47]. Previous and current use of tobacco and alcohol among adolescents and adults (parents and teachers) will be assessed using WHO's GSHS and STEPS tool guidelines, respectively [49, 50].

Standard equipment and procedures will be used for anthropometric, physiological and biochemical measures. Standard operating procedures will be developed to maintain uniformity and standardisation in sample collection and analysis.

For anthropometric measurements, ' 'UNICEF's standardised anthropometer will be used to measure the height to the nearest 0.1 cm. The portable electronic weighing scale will be used to measure the weight to the nearest 0.1 kg with minimum clothing and no shoes. Similarly, hip circumference (HC), waist circumference (WC), and Mid Upper Arm Circumference (MUAC) will be measured to the nearest 0.1 cm by using a fibreglass measuring tape. Further, triceps Skinfold Thickness (TST) and Sub Scapular Skinfold Thickness (SSFT) will be measured to the nearest 0.1 mm using Lange's skinfold calliper. Hand Dynamometer will be used to measure the maximum grip strength (average of three readings with a gap of one minute between each reading). Blood pressure (average of three readings) will be recorded for all participants using a digital sphygmomanometer (Omron). These instruments will be regularly calibrated as per standard requirements. The portable digital weighing scale will be checked for needle at '''zero' and calibrated after every 20th reading. ' 'Lange's skinfold calliper will also be checked for needle at '''zero' before every reading, and it will be calibrated using calibration block after every 20th reading.

Similarly, the hand dynamometer and digital sphygmomanometer will also be calibrated by following the instruction in their standard manual after every 100th reading. Also, fibreglass tape will be used for measuring hip, waist and mid-upper-arm-circumference as it is resilient to any wear and tear and does not expand in warm weather conditions as an ordinary measuring tape. Body Mass Index (BMI) will be calculated from the formula: weight (kg) divided by height ($m^2$). The calculated BMI will be categorised into underweight, normal, overweight, and obese as per the World Health Organization (WHO) cut-offs for adults (parents and teachers). For adolescents, their age and the gender-specific WHO growth reference chart will be used to categorise their BMI into severe thinness, thinness, normal, overweight, and obese.

For biochemical measures, blood samples (2 ml in Fluoride vial and 3 ml plain vial) will be obtained to estimate plasma glucose (mg/dL), total serum cholesterol (mg/dL), high density lipoprotein (mg/dL), low density lipo-protein (mg/dL) and triglycerides (mg/dL). Twenty-four-hour-urine and spot urine sample also will be collected to estimate the level of urinary sodium excretion (mg/day).

Explicit instruction on fasting will be given to all participants a day before collecting fasting blood samples. Similarly, they will also receive instructions regarding the process of 24-hour urine and spot urine collection. And urine collection containers will also be provided to them. They will be instructed to discard the first void of the morning and then start collecting the subsequent voids for the next 24 hours. They will also be advised to note down the time of their voids and whether they have collected each void in the collection container or not for the 24 hours period using a structured proforma. Blood samples for fasting plasma glucose and lipid profile and 24-hour and spot urine samples for urinary sodium excretion (mg/day), potassium, and chloride will be analysed using standard laboratory methods. Standard cut-offs will be used for adolescents and adults, respectively.

A team of three members, including the first author (SK) and two field investigators, will be involved in data collection. The field investigators will be qualified up to the graduation level and receive training for data collection. One of the investigators will be a laboratory technician for blood collection. Neither the investigator nor the data collectors will be blinded to the intervention.

After obtaining written informed consent of the study participants, questionnaires will be provided to them for writing their responses to questions in the classroom setting after that, anthropometric and blood pressure measurements will be taken. Finally, instructions for the

collection of blood and urine samples will be provided. Specific days will be fixed for the collection of blood samples in the school setting and urine samples from the home setting.

Before the final data collection, pilot testing will be undertaken to assess the adequacy of study tools and test the feasibility of implementing study protocol in school settings. A senior faculty member working in health promotion will do quality control of the data collection process. In addition, a Doctoral Committee comprising of seven faculty members of the institute, working independent from the sponsor and competing interests, has been explicitly constituted for data monitoring and progress evaluation of this project. The first author (SK) will be the implementer and evaluator of the intervention.

## Statistical analysis

The quantitative data analysis will be performed using the Statistical Package for Social Sciences (SPSS) version 21 based on intention-to-treat analysis. The complexity of cluster sampling design will be taken into consideration during the analysis. Descriptive statistical analysis will include the sample mean (along with standard deviation) and proportions according to the variable types.

ANOVA and Chi-square test will be used to compare the statistical significance of continuous and categorical variables. Within-group changes in quantitative variables from baseline to end-line at six months will be analysed using a paired t-test. Unpaired t-tests will be applied for assessing between-group comparisons. Multivariable regression analysis will be used to explore potential differences between groups (e.g., socio-economic status, age, and gender). The linear mixed-effects models will be used to estimate Difference-in-Differences (DiD) to assess the net effect of the intervention on primary and secondary outcome measures among adolescents. The net changes in primary and secondary outcome measures among parents and teachers will be considered as exploratory analysis [43]. Intention to treat analysis will be considered for the correlation between measures and loss of follow-up [51].

## Ethical considerations

Ethical permission has been obtained from the Post-Graduate Institute of Medical Education and Research, Chandigarh (INT/IEC/2018/000082; Date: 22/ 01/2019). Permission has already been granted for the selection of schools by the Department of Education, Chandigarh. Written assent of adolescents and consent for teachers and parents will be obtained after informing them about the study purpose, data to be collected, estimated time for data collection, the confidentiality of the data, and risks involved, before their enrolment in the study. As this is a behavioural intervention study, it involves no potential harm to the study participants. The reports for fasting plasma glucose, lipid profile, urinary sodium excretion, potassium and chloride levels will be given to the participants. Participants with blood pressure above 140/90 mmHg and abnormal blood or urine test reports in both the study arms (intervention and control) will be referred to the local health centre's doctors. Any modifications in the protocol will be conveyed to the institute ethics Committee after approval from the Doctoral Committee. These changes will be regularly updated on the Clinical Trial Registry-India. No later than three years after collecting the 1-year end-line assessment, we will deliver a wholly deidentified data set to an appropriate data archive for sharing purposes.

## Discussion

Behavioural risk factors for chronic diseases, such as unhealthy dietary habits, low physical activity levels, and tobacco and alcohol consumption, develop during early adolescence [2]. It is essential to 'catch them young' to prevent the development of these risk factors into various

intermediate conditions, including obesity, high blood pressure, elevated blood lipids and glucose levels, which can lead to various chronic diseases in adulthood [52]. If health promotion interventions are carried out in school settings, the effects could diffuse to many people and prove beneficial at a population level [53].

The use of health promotion interventions involving various interactive educational activities can effectively change individuals' behaviour [54]. However, there are several limitations with previous studies. Most of the previous studies have targeted individuals at high risk of disease, as they are more motivated than the general population [55]. Secondly, the control group did not receive any intervention in some trials, raising questions on the study's validity [56]. Thirdly, most studies did not have a theoretical framework to support the health promotion intervention [15, 16, 57]. Lastly, those studies conducted at the school setting level had explicitly focused only on school-going children. They did not look into the effect of the intervention on the behaviours of teachers or their parents [9, 12, 58]. The findings of our study will reveal the impact of health promotion interventions implemented in a school setting, not only on school children but also on their teachers and parents.

## Methodological considerations

The ability to measure any change requires the use of robust assessment methods. The 24-hour dietary recall methods have emerged as a reliable and helpful tool for dietary intake measurement compared to other tools, such as the Food Frequency Questionnaire (FFQ) [45]. We will use 24-hour dietary recall on two non-consecutive days as it has been shown to give a precise picture of individual dietary habits when taken on two or more times on non-consecutive days. The use of the 24-hour dietary recall tool became necessary in Indian settings as it imposes less burden on the participants when compared to other nutritional assessment tools. One of the limitations of using 24-dietary recall is underestimating the salt (sodium) intake [59]. Additionally, the salt intake will also be estimated through urinary sodium excretion from 24-hour urine samples collected for all the participants at the baseline survey, which is considered as the gold standard for evaluating salt intake [60–62]. In addition, urinary sodium excretion will also be assessed for all the participants through spot urine samples collected at the baseline and the end-line survey.

## Strengths and limitations

Before the final data collection, pilot testing will be undertaken to test the protocol's feasibility and assess research instruments adequacy. Pilot testing will also aid in identifying problems that could arise later in data collection [63]. Cluster RCT design has been chosen over an individual RCT to minimise the threat of any intervention contamination. The use of the randomisation process throughout the sampling procedure helps in minimisation of any potential selection bias. Another strength of the study is using previously validated and pretested tools in the Indian scenario, standard apparatus, and methodological guidelines of data collection for the anthropometric, physiological, and biochemical measures. Difference in difference (DID) method will help assess the intervention's true effect by not underestimating or overestimating the effect due to factors other than the intervention. Formative research, community consultation, and PRECEDE-PROCEED model for planning, implementation, and evaluation would help design a pragmatic intervention with potential for scale-up if found effective. One of the potential limitations of the study is the non-inclusion of private sector schools. The significant implication of excluding private schools is that it may lead to a lack of representation of participants across various socio-economic groups, limiting generalizability.

## Conclusion

Chronic diseases are on the rise. Hence, researchers, policymakers, and program implementers need to focus on evidence-based strategies for preventing the emergence of risk factors. Health promotion interventions implemented in the school setting simultaneously targeting adolescents, teachers, and parents could be more cost-effective. The PRECEDE-PROCEED model for planning, implementation, and evaluation of the intervention in a cluster Randomized Controlled Trial design with DID analysis could objectively assess the impact of scaling up primordial prevention to stem the tide of chronic diseases.

## Supporting information

**S1 Checklist. SPIRIT checklist.**
(DOC)

**S1 File. Institute's ethics committee approved protocol.**
(DOCX)

## Acknowledgments

We wish to extend our special thanks to Dr. Jacqui Webster, PhD, RPHNutri, Head, Public Health Advocacy and Policy Impact, Food Policy, Centre Director WHO Collaborating Centre Salt Reduction, Professor, Faculty of Medicine, UNSW, Sydney, The George Institute for Global Health, Australia for her technical inputs in improving the methodology and for reviewing the manuscript. We also thank the Director of School Education, Chandigarh Administration, for granting the permission to carry out this PhD project in public schools of Chandigarh Union Territory.

## Author Contributions

**Conceptualization:** Sandeep Kaur.

**Methodology:** Sandeep Kaur.

**Project administration:** Manmeet Kaur, Rajesh Kumar.

**Supervision:** Manmeet Kaur, Rajesh Kumar.

**Validation:** Manmeet Kaur, Rajesh Kumar.

**Writing – original draft:** Sandeep Kaur.

**Writing – review & editing:** Sandeep Kaur, Manmeet Kaur, Rajesh Kumar.

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
