## [Decision Letter · Decision Letter 0]

19 Aug 2021

PONE-D-21-16936

Health Promotion Intervention to Prevent Risk Factors of Chronic Diseases: Protocol for a Cluster Randomized Controlled Trial among Adolescents in School Settings of Chandigarh (India)

PLOS ONE

Dear Dr. Kaur,

Thank you for submitting your manuscript to PLOS ONE. After careful consideration, we feel that it has merit but does not fully meet PLOS ONE’s publication criteria as it currently stands. Therefore, we invite you to submit a revised version of the manuscript that addresses the points raised during the review process.

We look forward to receiving your revised manuscript.

Kind regards,

Rosely Sichieri

Academic Editor

PLOS ONE

Journal Requirements:

Additional Editor Comments (if provided):

I think the MS is appropriate for the journal.

Major issue is the lack of a clear objective of the study "Primary outcomes will be change in prevalence of behavioral risk factors from pre- to post-intervention. Changes in anthropometric, physiological, and biochemical measures will be the secondary outcomes."

The objective should be translated into a hypothesis to be tested.

Reviewers' comments:

Reviewer's Responses to Questions

**Comments to the Author**

1. Does the manuscript provide a valid rationale for the proposed study, with clearly identified and justified research questions?

Reviewer #1: Yes

Reviewer #2: Yes

2. Is the protocol technically sound and planned in a manner that will lead to a meaningful outcome and allow testing the stated hypotheses?

Reviewer #1: Partly

Reviewer #2: Yes

3. Is the methodology feasible and described in sufficient detail to allow the work to be replicable?

Reviewer #1: No

Reviewer #2: Yes

4. Have the authors described where all data underlying the findings will be made available when the study is complete?

Reviewer #1: Yes

Reviewer #2: No

5. Is the manuscript presented in an intelligible fashion and written in standard English?

Reviewer #1: Yes

Reviewer #2: No

6. Review Comments to the Author

You may also provide optional suggestions and comments to authors that they might find helpful in planning their study.

Reviewer #1: This paper represents an interesting study protocol that will investigate the impact of a school-based health promotion intervention on the risk factors of chronic diseases in students of 8th grade, their parents and teachers.

I have some questions.

Major issues:

1. The choose of primary and secondary outcomes are not instinctive. It would be more appropriate if the authors designed the study to impact on anthropometric, physiological, and biochemical measures, as behavioural risk factors could be considered as measures of adherence to the intervention.

2. Lines 134-136: The authors mention that the selection of the guardian who will participate to the study will be based on the choice of the adolescent. Wouldn't it be more appropriate for the guardians to decide who will participate, based on the time available to do so? Another question related to this: how do the authors intend to encourage parental adherence?

3. Lines 179-182: I understand that all administrators of the 24 pre-selected schools will receive a consent form to participate in the study, but only the first 12 that respond will participate. That being correct, what do the authors intend to do with the other possible positive responses that may arise? I suggest that only 12 consent forms are sent initially and for each refusal that arises, one more is sent.

4. Lines 238-239: As the activities will be aimed at all students enrolled in the 8th grade class, which will include adolescents with and without chronic diseases, I do not consider appropriate to encourage the use of medicine, only behavioral changes.

5. Lines 312-314: It is not clear for me why the authors are planning to use the Difference-in-Differences (DID) method. According to the cited reference (Wing et al., 2018), DID is a quasi-experimental research design often used to study causal relationships in public health settings where RCTs are infeasible or unethical, which is not the case.

6. Lines 385-425: It is important to use statistical analysis that consider the correlation between measures and the loss of follow-up

Minor issues:

1. Lines 96-107: I suggest that the authors keep only the general objective of the study, since the specific objectives reported represent steps to be taken to achieve the general objective of the study.

2. Lines 120-123: Please clarify how the intervention and control clusters will cater to a similar socio-economic group. Will the randomization be conducted in blocks?

3. Lines 214-217: Please describe in more details how the survey will be conducted. Will more participants be included? Or just the same ones from the intervention? If they are the same, it is considered a baseline data collection.

4. Lines 306-310: How will these compliance measures be recorded?

5. Lines 316-318: Could the authors please clarify when the 24-hour dietary recall will be collected? Moreover, it will be collected using any software? It will follow the AM-PM method?

Reviewer #2: I would like to start by complimenting the authors on this study. It is not easy to implement school-based interventions, and it is an exceptional challenge to conduct a large intervention such as the one documented in the present study in a developing country such as India. Although the authors have done a nice job, some issues concerning the introduction and design of the study should be highlighted.

Introduction

1. The introduction section has a logical flow yet lacks sufficient detail on existing literature. Although the majority of intervention trials focused on health promotion in the school setting have been conducted in developed countries, many others from developing one have already been published. Including this literature, is necessary to show what has been established, the knowledge gap(s). I suggest the authors balancing the introduction section with studies from similar economic contexts.

2. What do the authors mean by “Moreover, behaviour change interventions are often based on the perspectives of the researcher”? This sentence is not clear to me.

3. The fact that few studies in India have used theoretical frameworks and also few studies have been conducted among adolescents in the school settings of India is not a justification to conduct this study. Are there few intervention studies conducted in developing countries? The focus should not be on India but on low-to-middle income countries or other similar contexts. The authors need to make clear the gaps identified in the literature and the justification for undertaking the trial.

4. What are the hypotheses of the study?

Material and methods

1. In the “study área” section, the authors need to describe how the schools work and the students’ routine.

2. Lines 121-123. The authors should describe in detail how the randomization process was performed (e.g. type of randomization). Also, why did the authors perform the baseline assessment before the randomization?

3. In the “Inclusion and exclusion criteria” section, the authors should also describe the eligibility criteria for the study centres.

4. Line 248 change “Table 3” to “Table 2”.

5. Line 288. How many days per week of interactive learning?

6. Which strategies to improve adherence to intervention protocol will be adopted? And the procedures for monitoring adherence?

7. Line 317. The dietary behaviour assessment should be described in detail. Is the questionnaire self-administered? Interview? Who will conduct the interview? Multiple pass method? Weekdays or weekends?

8. Line 319. Remove “For adolescents”.

9. Is not clear to me who will generate the allocation sequence, who will enroll participants, and who will assign participants to interventions.

10. The statistical analysis to evaluate the effect of the intervention on the rate of change of the outcomes between groups is not adequate. Linear mixed-effect models should be considered by the authors.

11. In my opinion, the authors could remove “MET is the ratio of a person’s working

20 metabolic rate relative to the resting metabolic rate. One MET is defined as energy cost of sitting quietly and is equivalent to caloric consumption of one Kcal/Kg/Hour”.

12. Lines 402 to 425 should be included in the data collection section, according to the respective themes.

13. Figures resolution is low, but may be an issue with the compilation in the upload.

14. Finally, I suggest a final revision for English grammar and word choice.

7. PLOS authors have the option to publish the peer review history of their article (what does this mean?). If published, this will include your full peer review and any attached files.

Reviewer #1: No

Reviewer #2: **Yes: **Vitor Barreto Paravidino

---

## [Author Response · Author response to Decision Letter 0]

26 Sep 2021

RESPONSE TO EDITOR AND REVIEWER’S COMMENTS

Response to The Editor's Comments

Editor Comment: I think the MS is appropriate for the journal. Major issue is the lack of a clear objective of the study "Primary outcomes will be change in prevalence of behavioral risk factors from pre- to post-intervention. Changes in anthropometric, physiological, and biochemical measures will be the secondary outcomes."

The objective should be translated into a hypothesis to be tested.

Response: We thank the editor for the feedback and for coordinating the review process. We have revised the manuscript and addressed Reviewers’ comments. We have incorporated the suggestion and provided the hypothesis on the page no. five (Lines 99-110). 

General Comment: Please ensure that your manuscript meets PLOS ONE's style requirements, including those for file naming. 

Response: We have checked and attest that all formatting and style requirements have been met.

Response to Comments of Reviewer 1

We thank the reviewers for their time to review the manuscript. We agree to all the issues that have been raised. The response to each of the comments is provided in blue colour. The changes have been made in the manuscript using the ‘track changes’ command. 

This paper represents an interesting study protocol that will investigate the impact of a school-based health promotion intervention on the risk factors of chronic diseases in students of 8th grade, their parents and teachers.

Thank you for your encouraging words.

Comments and responses

1. The choose of primary and secondary outcomes are not instinctive. It would be more appropriate if the authors designed the study to impact on anthropometric, physiological, and biochemical measures, as behavioural risk factors could be considered as measures of adherence to the intervention.

We thank the reviewer for the suggestion and agree that the behavioural risk factors can be considered for measures of adherence to the intervention. Anthropometric, physiological and biochemical measures do not change in the short span of six months of intervention. While, behaviours do change. For the same reason, anthropometric, physiological and biochemical measures were considered as the secondary outcomes for the trial. Accordingly, we have classified these measures in the manuscript (page no. 10, lines 207-214) and added a reference (Reference no. 29 and 30). 

2. Lines 134-136: The authors mention that the selection of the guardian who will participate to the study will be based on the choice of the adolescent. Wouldn't it be more appropriate for the guardians to decide who will participate, based on the time available to do so? Another question related to this: how do the authors intend to encourage parental adherence?

Your suggestion is well taken, though we would ask the adolescents about the availability or time of the parent for the trial but ultimately parents would decide who will participate, we had missed it while writing. It has been clarified now at page no. 7 (Lines 145-147).

3. Lines 179-182: I understand that all administrators of the 24 pre-selected schools will receive a consent form to participate in the study, but only the first 12 that respond will participate. That being correct, what do the authors intend to do with the other possible positive responses that may arise? I suggest that only 12 consent forms are sent initially and for each refusal that arises, one more is sent.

We surely intended the same. The list of 24 randomly selected schools will be followed in sequence. For every refusal, the next school on the list will be approached. The first twelve schools from the list, consenting to be part of the trial will be enrolled in the study. Now, it has been clarified in detail at the page no. 9 (Lines 192-195).

4. Lines 238-239: As the activities will be aimed at all students enrolled in the 8th grade class, which will include adolescents with and without chronic diseases, I do not consider appropriate to encourage the use of medicine, only behavioral changes.

We thank the reviewer for the comment. We would like to explain that during intervention, the primary focus will be on the four risk factors, i.e., diet, physical activity, tobacco and alcohol use. The medicine adherence will be in only one session in schools from the intervention arm. The session's focus will be to help family member if anyone having any chronic diseases, such as diabetes and hypertension, know the importance of adhering to the medicine for the management and in case anyone develops these diseases later, they know the importance of adherence to prescribed medicines. It is explained on the page no.11 and 12 (Lines 248-254). 

5. Lines 312-314: It is not clear for me why the authors are planning to use the Difference-in-Differences (DID) method. According to the cited reference (Wing et al., 2018), DID is a quasi-experimental research design often used to study causal relationships in public health settings where RCTs are infeasible or unethical, which is not the case.

With thanks to the reviewer for this vital comment, we would like to clarify that it is evident from the literature that there may be effects of confounders even in the Cluster Randomized Controlled Trials (cRCTs) particularly when number of clusters are small. We have only 6 clusters in intervention and 6 clusters in control arm. As mentioned in the earlier cited reference, the DID design aims to adjust for hidden confounders to eliminate biases from the group- or time-invariant factors. So, to adjust any of the hidden confounders between the control and intervention arms, we have planned to use the Difference-in-Differences (DID) method for the present Randomized Controlled Trial. Please see additional reference no. 42 that clarifies the use of DID in the present study. We will also present results of intervention and control arm as is done traditionally in the analysis of cRCTs. 

6. Lines 385-425: It is important to use statistical analysis that consider the correlation between measures and the loss of follow-up

Yes, we agree and thank the reviewer for the suggestion. We have now added the details of the statistical analysis for correlation between measures and the loss of follow up at the page no. 21 (Lines 474-475).

7. Lines 96-107: I suggest that the authors keep only the general objective of the study, since the specific objectives reported represent steps to be taken to achieve the general objective of the study.

We agree and thank the reviewer for pointing this out. As the comment suggests, we have revised the specific objectives to general hypothesis to be testes with details given at page no. 5 (Lines 99-110).

8. Reviewer comment 8: Lines 120-123: Please clarify how the intervention and control clusters will cater to a similar socio-economic group. Will the randomization be conducted in blocks?

Yes, randomisation will be conducted in the blocks. We would like to clarify that after the baseline assessment of the trial, the income quartiles will be calculated based on information received from parents. Then average income of all twelve clusters will be calculated. This average income of the clusters will be used to categorize them into the different income quartiles. Then clusters having similar socio-economic status would be grouped in blocks. Lastly, half of the clusters from each income quartile group will be randomly allocated to the intervention arm and half to the control arm. It has been further clarified in Figure 1 at page no. 7 and lines 125-134. 

9. Lines 214-217: Please describe in more details how the survey will be conducted. Will more participants be included? Or just the same ones from the intervention? If they are the same, it is considered a baseline data collection.

After the intervention implementation, end-line survey will be carried in both arms, on the same study participants who will be included in the base-line survey using the same methods, tools, and apparatus as will be done in the baseline survey. Same has been mentioned in lines 230-233. 

10. Lines 306-310: How will these compliance measures be recorded?

The process of compliance measurement to be used has been explained at page no. 16 (lines: 354-361). A mid-intervention process evaluation will be carried out for measuring compliances. The mid-intervention process evaluation will provide information regarding the number of participants who attended various educational sessions, consideration about usefulness of intervention, reports of changes incorporated in behaviours, and the number of adolescents conveying the newly acquired information to their family members. These data will be recorded by interview of the adolescents in the intervention arm. 

11. Lines 316-318: Could the authors please clarify when the 24-hour dietary recall will be collected? Moreover, it will be collected using any software? It will follow the AM-PM method?

Thanks to the reviewer for seeking clarification on this important aspect of the dietary data collection. The first 24-hour dietary recall collection will be during the first interaction with all the participants in the schools. The second 24-hour dietary recall collection will be a week later. As the data collection of the 24-hour dietary recall will be manual and recorded in writing, so the Multiple-Pass Method is used and not the Automated Multiple Pass Method (AM PM). Later, the data will be entered in software (Excel sheet) for further analysis. These details have been added on page no. 17 (lines: 369-383).

Response to Reviewer 2

The authors would like to thank the reviewer for his comments that helped to improve the manuscript. We have addressed all the comments. The response to each of the comment in provided in blue colour. The changes have been made in the manuscript using the ‘track changes’ command. 

I would like to start by complimenting the authors on this study. It is not easy to implement school-based interventions, and it is an exceptional challenge to conduct a large intervention such as the one documented in the present study in a developing country such as India. Although the authors have done a nice job, some issues concerning the introduction and design of the study should be highlighted.

Thank you for your appreciation of the task at hand and keeping our moral high. We have tried our best to answer the issues raised by the reviewer from the very best of our understanding and knowledge.

1. The introduction section has a logical flow yet lacks sufficient detail on existing literature. Although the majority of intervention trials focused on health promotion in the school setting have been conducted in developed countries, many others from developing one have already been published. Including this literature, is necessary to show what has been established, the knowledge gap(s). I suggest the authors balancing the introduction section with studies from similar economic contexts.

We agree and have made changes following the suggestions at page no. 4 and 5 (lines:82-98). We have revised the introduction section and highlighted the gaps arising from similar studies from India and other low-middle income countries.

2. What do the authors mean by “Moreover, behaviour change interventions are often based on the perspectives of the researcher”? This sentence is not clear to me.

We would like to clarify that what we meant was that most of the previous studies have not used theoretical framework for intervention development and implementation. It is mentioned in the literature that interventions based on theoretical frameworks streamline the development of the intervention. 

We agree that the statement is not clear and does not convey the message clearly. We have removed the statement from the page no. 5.

3. The fact that few studies in India have used theoretical frameworks and also few studies have been conducted among adolescents in the school settings of India is not a justification to conduct this study. Are there few intervention studies conducted in developing countries? The focus should not be on India but on low-to-middle income countries or other similar contexts. The authors need to make clear the gaps identified in the literature and the justification for undertaking the trial.

We thank the reviewer for this insightful suggestion and agree that gaps in the literature need to be clearly mentioned in the 'Introduction' section to justify the study. We have identified gaps from the literature focusing on low-to-middle income countries and made changes at page no. 4 and 5(lines: 82-98). 

4. What are the hypotheses of the study?

Thanks to the reviewer for the comment. The null hypothesis of the study is: School-based health promotion intervention will not lead to any change in dietary intake of salt (g/day), sugar (g/day), fruits (g/day) and vegetables (g/day), physical inactivity (%), tobacco and alcohol use (%), anthropometric (height, weight, waist circumference, hip circumference, mid-upper arm circumference, subscapular and triceps skinfold thickness), physiological (blood pressure) and biochemical (fasting plasma glucose and urinary sodium excretion levels) measures among adolescents. 

The alternate hypothesis of the study is: School-based health promotion intervention will lead to change in dietary intake of salt (g/day), sugar (g/day), fruits (g/day) and vegetables (g/day), physical inactivity (%), tobacco and alcohol use (%) , anthropometric (height, weight, waist circumference, hip circumference, mid-upper arm circumference, subscapular and triceps skinfold thickness), physiological (blood pressure) and biochemical (fasting plasma glucose and urinary sodium excretion levels) measures among adolescents. It has been added at page no. (lines: 99-110).

5. In the “study área” section, the authors need to describe how the schools work and the students’ routine.

As suggested, we have added the adolescents' daily routine details under the 'Study Area' sub-section at page no. 6 (lines: 118-122). 

6. Lines 121-123. The authors should describe in detail how the randomization process was performed (e.g. type of randomization). Also, why did the authors perform the baseline assessment before the randomization?

The simple random sampling will be carried for selecting the twenty-four schools from the list of public schools provided on the administrative website using a random number table. Again, following the list of twenty-four schools, the first twelve schools giving consent to be part of the study will be selected. Later, as each school has four eight grade sections, namely, A, B, C and D. Cluster random sampling will be performed using random sampling method to select one section of grade 8th. All adolescents from the selected eighth-grade section will be eligible to participate in the study. 

The baseline assessment will be carried before the randomization to know the economic status of adolescent’s households. We would like to clarify that after the baseline assessment of the trial, the income quartiles will be calculated based on information received from parents. Then average income of all twelve clusters will be calculated. This average income of the clusters will be used to categorize them into the different income quartiles. Then clusters having similar socio-economic status would be grouped in blocks. Lastly, half of the clusters from each block will be randomly allocated to the intervention arm and half to the control arm. Thus, block randomisation will be done to select schools for intervention and control arm. As the number of clusters is small (6 in intervention and 6 in control arm), this method will be used to have similar economic status distribution in intervention and control arm. It has been further clarified at page no. 6 (lines:125-134).

7. In the “Inclusion and exclusion criteria” section, the authors should also describe the eligibility criteria for the study centres.

We agree and have now included the basis for including and excluding schools under the 'Inclusion and exclusion criteria' sub-section at page no. 7 (line:153). 

8. Line 248 change “Table 3” to “Table 2”.

We thank the reviewer for bringing this typing error to our notice. We have corrected it in the revised manuscript. 

9. Line 288. How many days per week of interactive learning?

There will be one interactive learning session for 30 minutes in one week for each of the intervention school for six months. This has been explained in detail in the ‘table 2’ given at page no. 15. 

10. Which strategies to improve adherence to intervention protocol will be adopted? And the procedures for monitoring adherence?

A micro-plan will be prepared ensure intervention adherence. It includes providing measuring tea-spoons of 5g to all the participants to promote the use of the recommended amount of salt and sugar in their daily routine. For physical activity, four outdoor game sessions of 30 minutes each will be carried with the help of the respective physical education teachers in all six schools every week. For tobacco and alcohol, peer-to-peer sessions will also be carried once every month to create awareness among the adolescents regarding the harmful effects of alcohol and tobacco consumption. It has been added at page no. 14 (lines: 302-308). To monitor adherence, mid-intervention process evaluation will be carried during intervention implementation. Information regarding the uptake of intervention, changes incorporated in behaviours, the number of adolescents following the behaviours and the number of them conveying the same to their family members will be collected in writing. This information will be recorded by interviewing adolescents in the intervention arm. 

11. Line 317. The dietary behaviour assessment should be described in detail. Is the questionnaire self-administered? Interview? Who will conduct the interview? Multiple pass method? Weekdays or weekends?

The dietary behaviour assessment will be through the Multiple-Pass Method by using two 24-hour dietary recalls on non-consecutive days during weekdays. After a thorough explanation of the dietary tool, all the adolescents in the same cluster will fill their forms themselves (self-administration). Whereas in-person interviews will be carried out with parents and teachers. The paper's first author will carry out all the interviews with parents and teachers and explain the tool to adolescents. The same has been added on the page number 17 (lines: 369-383).

12. Line 319. Remove “For adolescents”.

We thank the reviewer for bringing this typing error in notice. Extra words from the line have been removed. 

13. Is not clear to me who will generate the allocation sequence, who will enroll participants, and who will assign participants to interventions.

Second author (corresponding) will generate the allocation sequence and assign the schools and participants to the intervention arm. First Author will enrol participants. The same has been added at the page no. 6 (lines:132-134).

14. The statistical analysis to evaluate the effect of the intervention on the rate of change of the outcomes between groups is not adequate. Linear mixed-effect models should be considered by the authors.

Thank you! Yes, we agree and have added it at page no. 21 (lines:471-473) and will use it in analysis. 

15. In my opinion, the authors could remove “MET is the ratio of a person’s working

20 metabolic rate relative to the resting metabolic rate. One MET is defined as energy cost of sitting quietly and is equivalent to caloric consumption of one Kcal/Kg/Hour”.

We thank the reviewer for the suggestion and accordingly we have removed the above stated lines.

16. Lines 402 to 425 should be included in the data collection section, according to the respective themes.

We thank the reviewer for the input. We have included lines as per the themes in the ‘Data Collection’ section at page number 17 and 18 (lines: 368-392).

17. Figures resolution is low, but may be an issue with the compilation in the upload.

We have improved the resolution of the figures with the help of a professional as per the journal’s requirements. We have uploaded the revised figures with the improved resolution.

18. Finally, I suggest a final revision for English grammar and word choice.

Thank you! As suggested, we have revised the manuscript for the grammar and improved the terminologies.

---

## [Editor Report · Decision Letter 1]

16 Dec 2021

PONE-D-21-16936R1Health Promotion Intervention to Prevent Risk Factors of Chronic Diseases: Protocol for a Cluster Randomized Controlled Trial among Adolescents in School Settings of Chandigarh (India)PLOS ONE

Dear Dr. Kaur,

Thank for all changes in the new version,  I have 3 new comments, the most important about the design, sorry for not having included them before. None of the changes are required for acceptance. Sincerely,  Please submit your revised manuscript by Jan 30 2022 11:59PM. If you will need more time than this to complete your revisions, please reply to this message or contact the journal office at plosone@plos.org. Please include the following items when submitting your revised manuscript:A rebuttal letter that responds to each point raised by the academic editor and reviewer(s). You should upload this letter as a separate file labeled 'Response to Reviewers'.A marked-up copy of your manuscript that highlights changes made to the original version. You should upload this as a separate file labeled 'Revised Manuscript with Track Changes'.An unmarked version of your revised paper without tracked changes. You should upload this as a separate file labeled 'Manuscript'.If applicable, we recommend that you deposit your laboratory protocols in protocols.io to enhance the reproducibility of your results. Protocols.io assigns your protocol its own identifier (DOI) so that it can be cited independently in the future. For instructions see: https://journals.plos.org/plosone/s/submission-guidelines#loc-laboratory-protocols. Additionally, PLOS ONE offers an option for publishing peer-reviewed Lab Protocol articles, which describe protocols hosted on protocols.io. Read more information on sharing protocols at https://plos.org/protocols?utm_medium=editorial-email&utm_source=authorletters&utm_campaign=protocols.

We look forward to receiving your revised manuscript.

Kind regards,

Rosely Sichieri

Academic Editor

PLOS ONE

Additional Editor Comments:

1- low-middle income is the best definition for in developing countries

2- About the statement that there are almost no studies from middle-income countries: There are few studies conducted in Brazil as shown in the pooled analysis

School-based obesity interventions in the metropolitan area of Rio De Janeiro, Brazil: pooled analysis from five randomised studies.

Rodrigues RDRM, Hassan BK, Sgambato MR, Souza BDSN, Cunha DB, Pereira RA, Yokoo EM, Sichieri R.Br J Nutr. 2021 Nov 14;126(9):1373-1379. doi: 10.1017/S0007114521000076. Epub 2021 Jan 14.PMID: 33441203

3- I think it is still hard to understand the cluster allocation to treatment or control “ The average income of the clusters will be used to categorise them into different income quartiles. Then clusters having similar socio-economic status would be grouped in blocks. Lastly, half of the clusters from each block will be randomly allocated to the intervention arm and half to the control arm so that the intervention and control cluster caters to a similar socio-economic group”. Only 12 schools will be selected. At best it will be 2 schools per quintile. This is a paired intervention with necessary complex analysis. I would think that choosing a more homogenous population of schools and randomly allocating them to control or intervention would be a more adequate approach.
---

## [Author Response · Author response to Decision Letter 1]

20 Dec 2021

Response to The Academic Editor’s Comments

We thank the editor for the comments and for coordinating the review process. We have revised the manuscript and addressed the comments raised by the academic editor. The response to each of the comments is provided in blue colour. The changes have been made in the manuscript using the ‘track changes’ command. 

1. Low-middle income is the best definition for in developing countries

We agree with the editor’s comment and have changed the term developing countries with a more appropriate term, the ‘low- and middle-income’ countries

2. About the statement that there are almost no studies from middle-income countries: There are few studies conducted in Brazil as shown in the pooled analysis

School-based obesity interventions in the metropolitan area of Rio De Janeiro, Brazil: pooled analysis from five randomised studies.

Rodrigues RDRM, Hassan BK, Sgambato MR, Souza BDSN, Cunha DB, Pereira RA, Yokoo EM, Sichieri R.Br J Nutr. 2021 Nov 14;126(9):1373-1379. doi: 10.1017/S0007114521000076. Epub 2021 Jan 14.PMID: 33441203.

Thank you for bringing to our notice this recent publication (2021). We have added it in line 84 and cited the study with reference number 13. 

3. I think it is still hard to understand the cluster allocation to treatment or control “The average income of the clusters will be used to categorise them into different income quartiles. Then clusters having similar socio-economic status would be grouped in blocks. Lastly, half of the clusters from each block will be randomly allocated to the intervention arm and half to the control arm so that the intervention and control cluster caters to a similar socio-economic group”. Only 12 schools will be selected. At best it will be 2 schools per quintile. This is a paired intervention with necessary complex analysis. I would think that choosing a more homogenous population of schools and randomly allocating them to control or intervention would be a more adequate approach.

We thank the editor for providing us with a chance to clarify this important aspect of sampling design. 

There are 115 public schools in the Union Territory of Chandigarh. Data regarding socio-economic variables of students’ households in these 115 schools are not available for the selection of homogenous clusters. Primary data collection on these variables in all 115 public schools for selecting homogenous clusters will be expensive and time-consuming. 

As the number of clusters to be randomised in both arms are small, i.e., 12 clusters, in random selection, the distribution of socio-economic variables among the clusters selected in the two arms of the study may not be similar, which may confound the trial’s outcome. Hence, in view of limited resources available for the study, we would collect data only in the randomly selected clusters of 12 schools out of the total 115 schools. The data on households economic status will be used to stratify the clusters. Then, using the stratified random sampling method, clusters will be chosen to have a similar distribution of confounding variables in intervention and control arm clusters. We agree that this is a complex sampling design, but in view of the limited resources available, we think the proposed sampling design is more suitable for our study. 

We have revised the manuscript to convey the sampling design and analysis in a better manner (lines 129 to 133 and 464) and have added a reference (no.24) in the manuscript that supports the use of stratified random sampling in cRCTs specially when the sample size is small.

---

## [Editor Report · Decision Letter 2]

24 Jan 2022

Health Promotion Intervention to Prevent Risk Factors of Chronic Diseases: Protocol for a Cluster Randomized Controlled Trial among Adolescents in School Settings of Chandigarh (India)

PONE-D-21-16936R2

Dear Dr. Kaur,

We’re pleased to inform you that your manuscript has been judged scientifically suitable for publication and will be formally accepted for publication once it meets all outstanding technical requirements.

Kind regards,

Rosely Sichieri

Academic Editor

PLOS ONE
---

## [Editor Report · Acceptance letter]

2 Feb 2022

PONE-D-21-16936R2 

Health Promotion Intervention to Prevent Risk Factors of Chronic Diseases: Protocol for a Cluster Randomized Controlled Trial among Adolescents in School Settings of Chandigarh (India) 

Dear Dr. Kaur:

I'm pleased to inform you that your manuscript has been deemed suitable for publication in PLOS ONE. Congratulations! Your manuscript is now with our production department. 

Kind regards, 

on behalf of

Dr. Rosely Sichieri 

Academic Editor

PLOS ONE